# Offshore Fish Farms: A Review of Standards and Guidelines for Design and Analysis

**Yun-Il Chu** [1], **Chien-Ming Wang** [2,3,*], **Hong Zhang** [1], **Nagi Abdussamie** [3,4], **Hassan Karampour** [1], **Dong-Sheng Jeng** [1], **Joerg Baumeister** [1] and **Per Arild Aland** [5]

1   School of Engineering and Built Environment, Griffith University, Gold Coast, QLD 4222, Australia
2   School of Civil Engineering, The University of Queensland, St Lucia, QLD 4072, Australia
3   Blue Economy Cooperative Research Centre, Launceston, TAS 7248, Australia
4   Australian Maritime College, University of Tasmania, Launceston, TAS 7248, Australia
5   DNV Offshore Classification Maritime Sector, 1363 Høvik, Norway
*   Correspondence: cm.wang@uq.edu.au

**Abstract:** While moving fish farms to offshore sites can be a more sustainable way to expand farmed fish production, the fish pens have to contend with a harsher environment. Thus, it is necessary to draw on offshore engineering competences for designing and analysing the offshore fish farming infrastructure. This paper reviews existing design and analysis guidance from maritime classification and national/international authorities that can be applicable for offshore fish farms. Based on the existing design guidelines, a review of design criteria for offshore fish farms under the following subtopics is provided: design life, design environmental loads, combining environmental loads, and miscellaneous load conditions. This review on the global performance analysis procedures and methods is presented based on practices used for neighbouring industries, such as offshore oil and gas and wind energy production, under the following subtopics: hydrostatic analysis, hydrodynamic analysis, and mooring system analysis with introducing theoretical background and modelling techniques. This paper also highlights limitations and cautions when using these design and analysis methods. Providing this comprehensive information, as well as commentary on their applications, will help engineers and designers to develop offshore fish farming infrastructure with confidence.

**Keywords:** offshore fish farms; offshore engineering; design criteria; global performance analysis

## 1. Introduction

With diminishing appropriate fish farming locations nearshore, due to contested sea space utilizations and stressed ecosystems from massive fish faeces and uneaten feed, fish farm operators have started to install their fish farming infrastructure at more exposed/offshore sites [1–4]. These offshore sites offer large water columns because of deeper waters that can aid in waste dispersion, provide more pristine water and cooler water temperatures, as well as better return on investment from a larger scale of fish production [5–7]. On the other hand, offshore fish farming requires the use of large and complex fish pens in order to withstand the energetic environment. Therefore, designing of offshore fish pens will rely heavily on offshore engineering competences, including reasonable design condition selections and rigorous global performance analyses, including hydrostatic, hydrodynamic, and mooring system assessments [8].

In recent years, many offshore fish pen designs have been proposed, and some have been built and deployed. They range from structurally advanced versions to submerged structures, and some have even integrated offshore fish pens with offshore renewable energy production devices. More information and pictures of the various offshore fish pen designs and recent developments can be found in a review paper written by Chu et al. [6], and a book chapter by Wang et al. [7]. However, a consensus has not been reached as to which of these new offshore design concepts will become globally accepted and give

confidence in sustainable and profitable offshore fish farming to the aquaculture operators. The selection of standardized offshore fish pen designs will be made easier if technical guidance and design standards for offshore fish farms are available. These standards and guidelines can create a path forward for efficiency gains and reduction of future investment costs. Moreover, offshore fish pen designs based on rationally supported and scientifically evidenced guidance can gain credit from clients for adoption, and provide long-term investments.

Many design guidelines, including technical guidance and industry standards, are available from various sources that have been used as a design basis for offshore fish farms. In general, they focus on engineering design criteria, analytical methods, and techniques for reasoning and decisions for developing fish farming infrastructure in offshore environments. This paper compiles collected information for designing and analysing offshore fish pens from an offshore engineering perspective, and provides a review of the following topics that are considered important for making key decisions, such as dimensions, functional criteria, and serviceability requirements:

- Design criteria for offshore fish pens
  - Design life.
  - Design environmental loads: wave, current, wind conditions.
  - Combining environmental loads: for designing floating units, net and supporting system, and mooring system.
  - Miscellaneous load conditions.
- Global performance analysis and assessment
  - Hydrostatic analysis.
  - Hydrodynamic analysis: frequency domain analysis, time domain analysis.
  - Mooring system analysis: quasi-static method, dynamic method, mooring assessment for intact and damage cases.

This paper first presents existing standards and guidelines that are applicable for designing and analysing offshore fish pens. This is followed by design criteria for offshore fish farming infrastructure. The next section of the paper provides comprehensive information on the design and analysis methods, as well as commentary on their applicability and limitations. The contents of this paper should be useful to aquaculture engineers and designers when developing offshore fish pens.

## 2. Applicable Design Standards and Technical Guidance

### 2.1. Maritime Classification Rules and Standards

When designing a new product, engineers and designers shall consider a set of criteria that can guide their design process. Within the aquaculture industry, when it develops a new fish pen design, an approval of the new design is necessary to gain credit from clients and ensure the commercialisation of the design. In general, the design approval can be conducted by accredited maritime classification societies in accordance with their certifying rules and standards for offshore fish farming installations. The design approval has several advantages for fish pen manufacturers. Primarily, it clearly demonstrates to potential customers that the fish pen meets strict design standards, construction, and workmanship. Furthermore, the approval from maritime classification societies covers most fields of marine technology that can aid and advise throughout a plan approval and manufacturing process. Class engineers and surveyors will verify the manufacturer's calculations and analysis results, and help to maintain the quality of the design and end product. Note that the entry of classification services for offshore fish farming infrastructure is considered optional. However, in order to obtain the design approval and product certification, relevant classification rules shall be strictly obeyed. In this review paper, the term "rule" will be used when referring to classification rules for the design guidance.

In more recent years, the maritime classification organisations Det Norske Veritas (DNV), American Bureau of Shipping (ABS), Bureau Veritas (BV), and Lloyd's Register

(LR) have introduced their certifying rules, and provided classification services for offshore fish farming installations. Among classification societies, ABS and DNV officially provide certification rules and standards for offshore fish farming installations, as listed below:

- ABS, Guide for building and classing, aquaculture installations [9].
- ABS-FPI, Rules for building and classing, floating production installations [10].
- ABS-OI, Rules for building and classing, facilities on offshore installations [11].
- ABS, Position mooring systems [12].
- DNV-RU-OU-0503, Rules for classification, offshore fish farming units and installations [13].
- DNV-ST-C502, Offshore concrete structures [14].
- DNV-OTG-24, Fish escape prevention from marine fish farms [15].
- DNV-OS-C101, Design of offshore steel structures, general—LRFD method [16].
- DNV-OS-C102, Offshore standard for structural design of offshore ships [17].
- DNV-OS-C103, Structural design of column stabilised units—LRFD method [18].
- DNV-OS-C106, Structural design of deep draught floating units—LRFD method [19].
- DNV-OS-C201, Offshore standard for structure design of offshore units—WSD method [20].
- DNV-RP-C205, Environmental conditions and environmental loads [21].
- DNV-RP-F205, Global performance analysis of deepwater floating structures [22].
- DNV-OS-E301, Offshore standard for position mooring [23].

Other maritime classification groups, such as BV and LR, are known to provide classification services for their fish farming clients based on their existing rules and standards. In particular, BV has "Rules for classification and certification of fish farms" [24], which is a general rule for fish farming installations in nearshore waters (where the significant wave height does not exceed 2 m). However, where severe environmental conditions are the nature of the fish farms with various fish farm types and sizes, the following listed rules are applicable:

- BV, Rules and regulations for the classification of steel ships [25]: Applicable to ship-shaped fish farms.
- BV, Rules for the classification of offshore units [26]: Applicable to semisubmersible fish farms or fish farms using techniques from the offshore field.

To the authors' best knowledge, LR's rule for offshore fish farming has not yet been published. Nevertheless, a book chapter written by Barker [27] presented LR's principles of approval for offshore fish pens, which take a similar approach to other maritime classification groups.

In spite of the existence of applicable rules, classification approvals do not guarantee all offshore fish farming elements. For example, the ABS provides technical guidance and class requirements for the following fish farming elements only:

- Fish farming hull structure.
- Mooring system and foundation.
- Onboard machinery, equipment, and systems that are not considered as aquaculture systems.
- Similarly, the DNV rule (DNV-RU-OU-0503) indicates a limited coverage, such as:
- Hull including superstructure.
- Crane pedestal.
- Attachments of helideck support structure.
- Structural interfaces between hull and components.
- Foundation and support for heavy equipment.

Note that the DNV rule covers mooring design for units when the units are intended to be moored during inservice.

Certification rules from ABS do not cover floating collars and net pens which are made of polymers, concrete, or equivalent, and their associated equipment, feeding and production facilities, feedstock facilities, and fish escape prevention devices. The coverage of DNV's certification rules is almost the same as that of ABS, except that DNV provides

technical guidance for offshore concrete structures [14] and fish escape prevention [15]. Both ABS and DNV rules indicate that noncovered elements shall be assessed under the jurisdiction of local aquaculture authorities. Moreover, classification rules are mainly intended to classify steel or metallic material structures, as listed in Table 1. Their rules do not explicitly address other types of fish farming units or installations that have design alternatives. Similarly, BV limits its classification rules by not covering structures, pressure vessels, machineries, and equipment if they are not related to the global safety of the fish farms. Nets are beyond the scope of classification, except for special concerns of significant induced loads on the fish farm structure. Figure 1 shows a diagram of coverage and noncoverage of classification rules based on the collected information.

**Table 1.** Classification types for offshore fish farms according to ABS, DNV, and BV.

| | ABS | DNV | BV |
|---|---|---|---|
| Types | • Spar-type<br>• Column-stabilized type<br>• Ship-shaped type<br>• Nonbuoyant type (i.e., bottom fixed) | • Deep draught type<br>• Column-stabilized type<br>• Ship shaped type<br>• Self-elevating type<br>• Cylindrical type | • Barge or converted ship<br>• Rigid pontoon<br>• Submersible cage unit<br>• Articulated unit<br>• Watertight cage unit<br>• Bottom-resting unit<br>• Flexible cage unit<br>• Modifiable cage unit<br>• Ballastable unit |

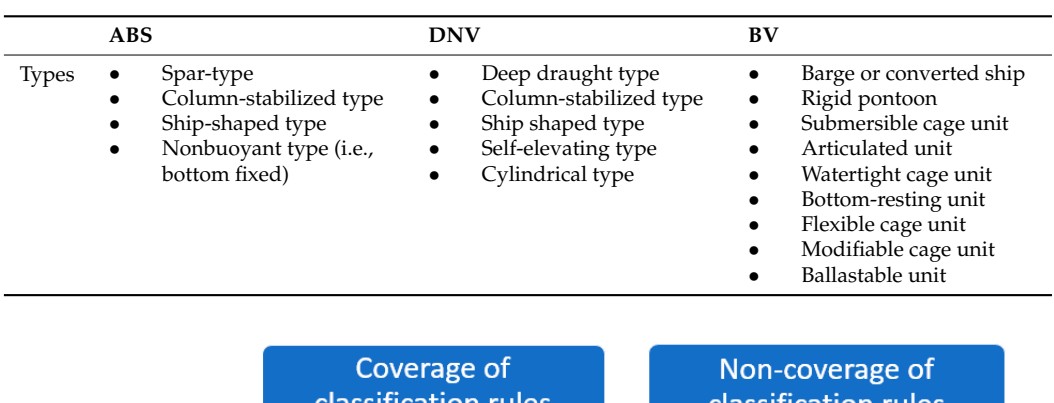
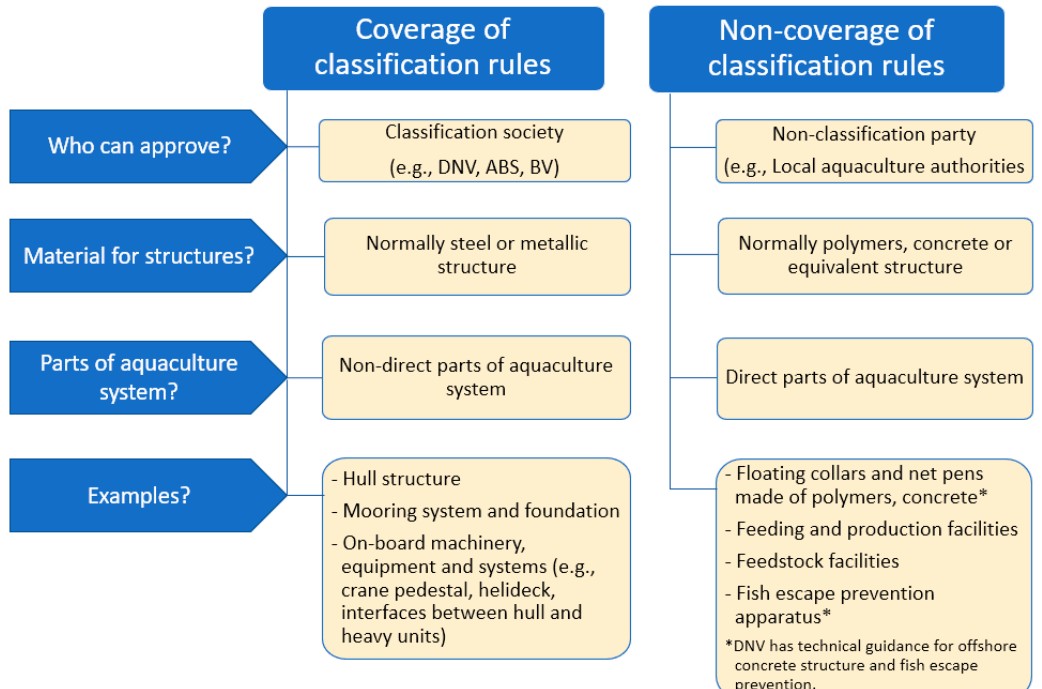

**Figure 1.** Diagram of coverage and noncoverage of classification rules for offshore fish farms.

Among classification types for offshore fish farms, there are already three major types: (i) ship-shaped, (ii) column-stabilized, and (iii) spar/deep draught, which are already built or being planned for construction. Note that tendon-based floating units, such as tension leg platforms (TLPs), have been used for fish farming in some countries. However, this type is not as popular as the major classification types mentioned above, and the type is not clearly indicated in the classification rules above. According to the literature review conducted by Chu et al. [6], TLP mooring systems work more like a fixed structure, and wave forces are directly counteracted with appropriate tendon stiffness. Therefore, TLPs may not be a suitable type for high volume mass fish pen design. In addition to this,

only BV has classification rules for submerged-type fish pens, but only when the pens are sited nearshore.

Based on the listed classification types, relevant design codes from DNV and ABS are presented in Table 2.

**Table 2.** DNV and ABS design codes of different classification types for offshore fish farming.

| Types | DNV's Design Codes | ABS's Design Codes |
|---|---|---|
| Ship-shaped | DNV-OS-C101, C102 | ABS-FPI Section 5A-1-4, 5A-1-5 |
| Column-stabilized | DNV-OS-C101, C103, C201 | ABS-FPI Section 5B-1-1, 5B-1-2 |
| Spar/Deep draught | DNV-OS-C101, C106 | ABS-FPI Section 5B-3-1, 5B-3-3, 5B-3-4 |

*2.2. National and International Standards*

In order to verify the entire design for offshore fish farm installations, it is necessary to include aquaculture elements which are not covered by the maritime classification societies. An alternative verification is necessary, with consideration of application of national governances or international standards, such as:

- Norway Standard, NS 9415, Marine fish farms requirements for design, dimensioning, production, installation, and operation (inclusive Norway Standard, NS 3472, Steel Structures—Design rules) [28].
- Marine Scotland, A technical standard for Scottish finfish aquaculture [29].
- ISO 16488:2015, Marine finfish farms—open net cage-design and operation [30].

The Norwegian standard NS 9415 was developed to specify technical requirements for dimensioning, design, installation, and operation of floating fish farming installations. This standard, the first of its kind internationally, was established in collaboration with representatives of industry, research institutions, and authorities. Although the NS 9415 mainly regulates nearshore fish farming activities (not for offshore fish farming activities), the basis of the technical requirements can be similarly considered, even at offshore sites, with a rational approach regarding harsher offshore environmental conditions when compared to nearshore sites.

The Scottish technical standard was developed by a group of experts to minimise fish escapes in Scotland. It was designed to fit proportionally for the Scottish finfish farming industry, and the standard was drafted with wide-ranging consultation and engagement of all stakeholders since 2010. This standard determines technical requirements for fish farm equipment. This standard applies to all species of finfish farming in Scotland. It should be used in conjunction with operational procedures, codes of practice, operator's manuals, and staff training to ensure equipment is properly used and maintained, and that operational procedures are followed correctly.

ISO 16488:2015 suggests a general method that can be used for the systematic analysis, design, and evaluation of net cage marine finfish farms. The methodology presented in this international standard allows assessing the suitability of floating structures, nets, and mooring equipment for a given finfish farm and its environment. This standard covers the specification of design criteria through assessment of environmental conditions and acceptable risks, and specifies acceptable techniques for the design and analysis of finfish farms. This international standard also provides guidelines for developing a handbook, by documenting procedures for the correct maintenance and operation of finfish farms. The application of this standard is aimed to reduce the risk of escape from marine finfish farms. It is designed to be used by operators of net cage marine finfish farms. Through the application of this international standard, it is intended to achieve increasing integrity levels of human safety systems [30].

The aforementioned standards cover scopes in assessing safety and integrity levels, so that designers can choose any of these to ensure their design's reliability. However, these standards shall be considered as volunteering application, depending on the jurisdiction of governing localities for offshore fish farms, since there might be insufficiencies to consider

harsher offshore environmental conditions. Therefore, this review paper will primarily focus on maritime classification rules for offshore fish farming, and supplementally review national and international finfish farming standards.

## 3. Design Criteria for Offshore Fish Farms

This section presents various aspects of the recommended design criteria referred from previously mentioned classification rules and standards, as well as regulations of international and local authorities. It starts with the expected design life, followed by design environmental loads that are recommended in designing offshore fish farm infrastructure, together with methods to compose such environmental loads for setting up the design load condition at the same probability level. Note that the design criteria recommended here shall be limited to designs for the classification entry (i.e., verified designs by classification groups) in compliance with the design specification, material selection and inspection, construction survey, and maintenance instructions required by the relevant maritime classification rules and standards.

### 3.1. Design Life

The design life of a component or product is how long the designer expects the item to operate within its specified parameters [31]. In other words, design life is the forecasted life of products based on their design, and it can be theoretically calculated based on expected conditions, uses, and physical properties of the products [32].

In general, design life for offshore fish farming infrastructure varies depending on their applicable design standards and guidelines, maintenance methods, manned or unmanned operations, risks associated with farm sites, marine warranty, and special requirements from operators. Nevertheless, establishing a design life is important for stable and long-term marine aquaculture growth, so that the fish farm operator can coordinate measurable and predictable regime covering issues related to structural integrity, as well as safety of the workers and fish. In addition, the design life is also directly linked to insurance liabilities. Insurance companies are increasingly curious about the potential of offshore fish farming as a niche area for business growth [33]. However, the offshore fish farming industry has not yet come up with appropriate insurable solutions to provide comprehensive coverage needed for the risks associated with offshore operations.

Nevertheless, the aforementioned classification rules can be adopted to specify the expected design life for offshore fish farming infrastructure. Those rules have been widely used for the steel construction of offshore oil and gas and wind industries, with sufficient grounds to ensure the serviceability of offshore structures for more than 20 years of design life. However, the application of classification rules should be limited under the premise that the same certifying scopes of design specification, material selection, equipment inspection, and construction survey are required for the offshore units, especially steel-built structures, for acquiring the same level of accredited design life.

In accordance with the ABS's guideline, any new offshore fish farming structure (regardless of types mentioned in Table 1) is targeted to have at least a 20 year design life for steel structures [9]. However, in accordance with ABS's rules for floating production installations [10] or acceptable equivalent criteria, when the design life is greater than 20 years and the offshore fish farming installations are designed without any dry-docking service, the nominal design corrosion values of the steel structure should be increased (i.e., the steel structure will corrode quickly). The DNV's basic design lifetime of ship-shaped and column-stabilized types for offshore fish farming installation indicates a minimum of 20 years, accounting for corrosion that is likely to occur during the design period in offshore environmental conditions [20].

Assumption of design life is important when measuring design environmental loads, as they link directly to the probability exceeding characteristics of the environmental loads. The design of offshore fish farming installations shall be based on the most severe environmental loads which the structure may experience during its design life. NS 9415

indicates that the design life of a floating collar should not be set to less than 10 years, and the return period of extreme loads should be at least 2.5 times the design operational life. This requirement is based on the probability of exceeding characteristic sizing of 2% probability per year [28]. For example, if the design run time is set to 20 years, this equates to a return period for extreme loads of 50 years.

*3.2. Design Environmental Loads*

Quantifying design environmental loads is one of the primary steps to have an optimal design for offshore fish pens. Both the operating and extreme environmental conditions shall be considered to represent the offshore candidate site [34].

In general, offshore fish farming structures are compliant in nature. It consists of flexible parts such as mooring chains, ropes, and fish nets. Moreover, in the offshore environment, waves, currents, and winds generate dynamic forces that induce interactive motions that can result in large displacements and rotations [35]. With these dynamic influences, it is important to estimate accurate or acceptable force distributions through analytical procedures. Therefore, a design guidance is needed to suggest a method to employ dynamic environmental loads and their combinations, called design environmental loads and conditions.

From various sources, methods to estimate and combine three major dynamic environmental loads, wind, wave, and current, are introduced herein.

3.2.1. Wave

According to the ABS rule, the wave forces acting on offshore fish farming installations may consider following major force components [10]:

(a)    First order forces at wave frequencies.
(b)    Second order forces at low frequencies.
(c)    Second order forces at high frequencies.
(d)    Mean drift force (steady component of the second order forces).

In order to calculate the aforenoted components, the fluid potential theory and the Morison equation are commonly used for solving the wave forces [36–39]. The three-dimensional (3D) panel method, based on the fluid potential theory, is the most common numerical approach used for analysing hydrodynamic responses of a large-volume structure in waves. The method can represent the structure surface by a series of diffraction panels. On the other hand, the Morison equation approach is widely used for slender body components. To have accurate wave forces on the complex offshore fish farming structures, a hybrid method to consider both large-volume surface components by 3D diffraction panels, and small cross-sectional components by Morison elements is commonly accepted by classification rules for offshore fish farming installations [10,22].

**(1)    First order wave force (Linear wave force)**

A linear potential wave analysis will usually suffice for prediction of the first order forces at wave frequencies [40–42]. The term linear means that the pressure of fluid dynamics and resulting loads are proportional to the wave amplitude. For calculation of the wave loads on large-volume structures, which significantly alter the incident wave field, the linear potential wave theory with the boundary element method is often used to account for both the incident wave force and the forces resulting from wave diffraction and radiation.

The fluid flow field near a floating body may be defined as a velocity potential [42], as:

$$\phi\left(\vec{X}, t\right) = a_w \cdot Re\left[\varphi\left(\vec{X}\right) e^{-i\omega t}\right] \tag{1}$$

in which $a_w$ is the incident wave amplitude, $Re[]$ is the real part of the argument, $\varphi\left(\vec{X}\right)$ is the complex space-dependent potential function, $\omega$ is the wave angular frequency, and

$\vec{X}$ is the position vector of a point on the floating body surface. The potential function is complex, but the resultant physical quantities, such as fluid pressure and body motions, in the frequency domain analysis can be obtained by considering the real part only.

In Equation (1), $\varphi\left(\vec{X}\right)$ may be distinguished from contributions of the incident waves, the diffracted waves, and the radiation waves induced by six degrees of freedom body motions [42]. Therefore, it may be written as:

$$\varphi\left(\vec{X}\right) = (\varphi_I + \varphi_d) + \sum_{k=1}^{6} \varphi_{rk} \tag{2}$$

where $\varphi_I$ is the first order incident wave potential with unit wave amplitude, $\varphi_d$ is the corresponding diffraction wave potential, and $\varphi_{rk}$ is the radiation wave potential due to a unit motion amplitude in the $k$-th motion.

Knowing the wave velocity potentials as a special case with unit amplitude $a_w = 1$, the first order hydrodynamic pressure distribution can be calculated by using the linearized Bernoulli's equation. In the case of pressure distribution, various fluid forces can be calculated by integrating the pressure over the wetted surface of the body [42].

Total first order forces $F_j$ can be written as:

$$F_j = \left(F_{Ij} + F_{dj}\right) + \sum_{k=1}^{6} F_{rjk} \tag{3}$$

where $j = 1$–6 represent the $j$-th normal vector of freedom, $F_{Ij}$ is the incident wave force, $F_{dj}$ is the diffracting force, $F_{rjk}$ is the radiation force induced by the structure's $k$-th degree of freedom rigid body motion.

On the other hand, for floating structures composed of slender members that do not significantly diffract the incident wave field (i.e., fully submerged), one may use semiempirical formulations, such as the Morison equation [42], given by:

$$F = \rho_w V\left(\dot{u} + C_A\left(\dot{u} - \dot{v}\right)\right) + \frac{1}{2}\rho \cdot C_D \cdot A \cdot (u - v)|u - v| \tag{4}$$

where $\rho_w$ is the water density, $\dot{u}$ is the flow acceleration, $\dot{v}$ is the acceleration of floating structures, $u$ is the flow velocity, $v$ is velocity of floating structures, $C_A$ is the added mass coefficient, $C_D$ is the drag coefficient, $V$ is the displaced volume of water, and $A$ is the cross-sectional area of the body perpendicular to the flow direction. The first term on the right side represents the inertia force, and the second term represents the drag force. In general, applications of the Morison equation may be used for slender structures with diameters less than one-fifth of the wave lengths.

In general, the column-stabilized type of offshore fish farms consists of several large columns and pontoons, and small cylindrical braces. A combination model of diffraction panel elements and Morison elements may be adopted for the calculation of hydrodynamic characteristics of the first order wave.

**(2)  Second order wave force at low frequency**

Low frequency motions of a moored fish farming installation can be severely affected by the slowly varying wave drift force, which is a second order wave force [22]. This wave force is proportional to the square of the wave amplitude [43]. Estimation of the second order wave effect is one of the major design and analysis concerns for performance and safety in floating structures for offshore fish farming.

The second order wave force in a random sea-state is normally represented by a sum of $N$ wave components $\omega_i$ and $\omega_j$, $(i, j = 1 \sim N)$, and this force oscillates at difference frequencies $\omega_i - \omega_j$, as given by the expression:

$$F^{(2)}(t) = Re \sum_{i,j}^{N} a_i a_j H^{(2)}\left(\omega_i, \omega_j\right) e^{i(\omega_i - \omega_j)t} \tag{5}$$

where $a_i$, $a_j$ are the individual wave amplitudes, superscript (2) is the second order variation, $H^{(2)}$ is the quadratic transfer function (QTF) to calculate the difference frequency loads, and Re denotes the real part. The QTF is presented as a complex quantity with amplitude and phase, which requires significant discretization. Therefore, commercial software packages (e.g., ANSYS AQWA [44], WADAM [45], NEMOH v.3 [46]) are often used for calculations [47]. Note that more details on the second order wave theory and solution at low frequency motions can be found in a paper written by Choi et al. [43].

In order to simplify the calculations (i.e., simplifying the full QTF matrix), Newman's approximation may be adopted, which is generally accepted for the hydrodynamic analysis of moored offshore structures in moderate and deep waters with long crested waves [22,48]. By using Newman's approximation, computational time is significantly reduced, as a linear analysis can be adopted. The off-diagonal elements in the full QTF matrix can be approximated by the diagonal elements, as:

$$H^{(2)}(\omega_i, \omega_j) \cong \frac{1}{2}\left[H^{(2)}(\omega_i, \omega_i) + H^{(2)}(\omega_j, \omega_j)\right] \tag{6}$$

As the diagonal elements of the QTF matrix can be calculated from the first order velocity potential, there is no need to calculate the second order velocity potential.

Newman's approximation usually gives satisfactory results for slow-drift motions in the horizontal plane when the natural period is much larger than the wave period. However, for slow-drift motions in the vertical plane (e.g., heave/pitch motions of spar type floaters), Newman's approximation may underestimate the slow drift forces, and in such case a solution of the full QTF matrix is required. In particular, for netted structures with new floater concepts and/or relatively shallow water installations, caution should be taken as to whether Newman's approximation is applicable [22,49].

**(3)  Second order wave force at high frequency**

Second order wave force in a random sea-state oscillating at the sum frequencies $\omega_i + \omega_j$ excites a vertical resonant response, in particular to the tendon stabilized floating units such as tension leg platforms (TLPs) [22]. Since stiff tendons have a relatively high resonant frequency, vertical vibration near the resonant frequency can be excited by second order wave forces. This high frequency vibration is called springing, and the springing responses should be monitored when TLPs are chosen as a main floating unit for fish farming.

In order to calculate the sum frequencies $\omega_i + \omega_j$, the quadratic transfer function (QTF) can be used as similar to the different frequency wave loads. The sum frequency force in a random sea-state can be expressed as:

$$F^{(2)}(t) = Re\sum_{i,j}^{N} a_i a_j H^{(2)}(\omega_i, \omega_j) e^{i(\omega_i - \omega_j)t} \tag{7}$$

**(4)  Mean drift force**

Based on the mean wetted body surface integration approach (i.e., near field method), general forms of the average wave drift force acting on a floating body in all directions of motions can be given as a special form from Equation (5), of which $\omega_i = \omega_j$ and the sum frequency force components are excluded [22,42], so as to be expressed as:

$$F^{(2)} = \sum_{i=1}^{N} a_i^2 Re\left[H^{(2)}(\omega_i, \omega_i)\right] \tag{8}$$

3.2.2. Current

The estimation of current loads is a challenging task due to different local topographic conditions that vary greatly in magnitude and direction with depth. Therefore, it is difficult to provide sufficient background for determination of design current speeds and directions. Nevertheless, current loads can be the dominating steady force on the slender structures,

mooring lines, and nets for offshore fish farming installations, and in particular to the fully submerged fish pen designs. It is therefore important to apply a reasonable current force with attention to the current excitation force, as well as the damping contribution on the related structural members [22].

Appropriate current forces are to be calculated in areas where relatively high velocity currents occur on the submerged structures, mooring lines, and nets. The current forces on the submerged parts may be calculated as the drag force term of the Morison equation, given by Equation (4). In order to find the applicable extreme current velocity, the current velocities can be statistically analysed as a probabilistic distribution, such as the Weibull distribution [29,34].

BV's rule for fish farms states that the current speed, direction, and velocity profiles are to be specified for circulational-, tidal-, and wind-generated components. Otherwise, the certification and classification will be based upon the following uniform current velocities at the still water level:

- Circulational + tidal components: $V_t(h) = 0.5$ m/s.
- Wind-induced component: $V_w(h) = 0.02 \times V_{10}$ m/s, where $V_{10}$ is the 10 min wind speed at 10 m above still water line.

According to NS 9415 [28], in order to set the current speed and estimate an extreme value for the specified return period, the multiplication factors given in Table 3 can be used to factor the maximum current speed over at least a four week measurement period at the site.

**Table 3.** Multiplication factors as a result of return period.

| Return Period (Year) | 1 | 10 | 50 | 100 |
|---|---|---|---|---|
| Multiplication factor | 1.40 | 1.65 | 1.85 | 2.00 |

For example, if the maximum current over four weeks is 0.4 m/s, the 10 year current is assumed to be $0.4 \times 1.65 = 0.66$ m/s, while the 50 year current is set as $0.4 \times 1.85 = 0.74$ m/s, and the 100 year current is set as $0.4 \times 2 = 0.8$ m/s.

With regard to the direction of the current, the technical standard for Scottish finfish aquaculture states that the design current shall be at least eight concurrent directions, including the direction aligned to the highest speed current, may be expected [29]. On the other hand, the Norwegian standard indicates that the current direction shall define the dominant direction of the fish farm installation. The most unfavourable direction that provides the highest load should be used, unless the installation is moored to the prevailing current direction [28].

When several net pens are arranged in a row, such as Havfarm 1 (see a review paper written by Wang et al. [7]), the rear net pen is less affected by the current force than the front net pen [50–53]. The current velocity drops rapidly after passing through nets one after the other in a row. This phenomenon is the result of the shielding effect, by which some of the water flow loses momentum when water flows through the front net. Owing to the loss of momentum, a continuing reduction of flow velocity occurs in the downstream fish pens. The current reduction factor, $R$ can be defined as:

$$R = 1 - \frac{u_i}{U_\infty} \tag{9}$$

where $U_\infty$ is the incoming current velocity, and $u_i$ is the local current velocity felt by the $i$-th net panel facing the flow.

In order to calculate the local velocities on the net panels, a simple engineering approach was suggested by Løland [54]

$$u_i = U_\infty \cdot (1 - R) = U_\infty \cdot (1 - r)^{i-1}$$
$$r = 0.46 \cdot C_D \tag{10}$$

where $r$ is the current velocity reduction factor behind one net panel. $C_D$ is the drag coefficient on the net panel which depends on the solidity ratio of the net panel. $C_D$ can be approximated by using Equation (11), suggested by Lader and Enerhaug [55], based on flexible net structure experiments.

$$C_D = 0.04 + \left(-0.04 + 0.33S_n + 6.54S_n{}^2 - 4.88S_n{}^3\right)\cos(\theta) \qquad (11)$$

where $S_n$ is the solidity ratio of the net panel and $\theta$ is the angle between relative inflow direction to the net normal vector. Solidity ratio ($S_n$) of the net panel is defined as the ratio between the projected area ($A_p$) covered by the threads and the total area of the net panel ($A$):

$$S_n = \frac{A_p}{A} \qquad (12)$$

Note that the given Equation (11) is an empirical approach that can be highly dependent on the experimental setup. Other methods can also be adopted, such as by utilizing the Morison equation to mesh line/bar elements [56] or screen type method by applying load to membrane net panels [57]. However, to the authors' knowledge, such approaches are not endorsed by maritime classifications, and the net is considered outside the classification scope.

### 3.2.3. Wind

Wind conditions can be important environmental parameters for predicting global motion responses of offshore fish farming installations. Especially, for the integrated system of offshore fish farms and wind turbines, the wind loads can be the dominating excitation force [41,58]. Therefore, accurate modelling of the wind effect is essential for the integrated designs with structural components above the water line. In general, the wind conditions are to be established from collected wind data at the relevant offshore site. The conditions should be consistent with other environmental parameters (e.g., wave, current) assumed to occur simultaneously [10]. The statistical distribution to calculate the extreme wind load is to be based on the analysis and interpretation of wind data by a recognized method. It includes the frequency distributions of wind velocity and direction, and the recurrence period of extreme winds.

The wind loads acting on floating structures consist of two components: a static part resulting in an average offset and average slope, and a dynamic part due to wind gusts that excite the low frequency motions in surge, sway, and yaw. Owing to its importance, the wind loads are usually determined by wind tunnel tests [59]. These tests are very often conducted early in the design stage. If significant changes are made to the deck/topside structures during detail design, repeating these wind tunnel tests may be needed. The gust wind loading component can be simulated by a wind gust spectrum, which can be adopted from existing wind spectra such as API and NPD spectra [22]. It should be emphasised that a wind spectrum should be selected that best represents the actual geographical area in which the installation is located.

According to the ABS rule for floating production installation [10], the wind load can be considered either as steady wind forces or a combination of steady and time-varying loads, which can be described as below:

- If wind force is considered as a steady force, the wind velocity based on 1 min average velocity is to be used to calculate the wind load.
- Effect of wind gust can be considered as a combination of steady load and a time-varying component calculated from an appropriate wind spectrum. In this approach, the wind velocity based on 1 h average velocity should be used for the steady wind load calculation.

Note that the former approach is preferred when the wind energy spectrum cannot be reliably derived [60].

In addition, the BV rule [24] states that for fish farms, 1 min wind velocity at 10 m above the mean water level is to be taken not less than 36 m/s for normal working and transit conditions, and not less than 51.5 m/s for the severe storm condition.

Based on the Norwegian standard [28], when designing main components and total facilities for fish farms, a 50 year return period wind shall be established. If no empirical wind data is available for the relevant location, 35 m/s must be used as the design wind load. If the wind data from meteorological stations is available, a 50 year return period shall be used for dimensioning loads. The maximum wind speed shall be indicated as 10 min of mean wind at a reference height of 10 metres above sea level. Measurements shall take place over a period of three months at a location, with subsequent statistical analysis and extrapolation.

According to DNV's recommendations, the averaging time for wind speeds and the reference height must always be specified, as the wind speed varies with time, and it also varies with respect to the height above the sea surface. A reference height of $H = 10$ m and average times of 1 min, 10 min, and 1 h are commonly adopted, and wind speed averaging over 1 min is often referred to as the sustained wind speed.

The shapes of structures exposed to wind and vertical height from the sea water surface will influence wind pressure on a particular windage (i.e., area exposed to wind). ABS suggests a simple equation that makes use of some coefficients to consider different shapes at different heights to calculate the wind pressure $P_{wind}$.

$$P_{wind} = 0.61 C_s C_h V_{ref}^2 \tag{13}$$

where $C_s$ is the shape coefficient, $C_h$ is the height coefficient, and $V_{ref}$ is the velocity of wind at a reference height of 10 m from the sea water surface for 1 h average speed. Examples of coefficients are given in Tables 4 and 5. More information can be found in ABS's guidance [10].

**Table 4.** Shape coefficient $C_s$ for windages.

| Shape | $C_s$ |
|---|---|
| Sphere | 0.4 |
| Cylindrical | 0.5 |

Note that ABS does not specify other shapes, but suggests applicable shape coefficients depending on parts of the structure.

**Table 5.** Height coefficient $C_h$ for windages.

| Height above Water Line (m) | $C_h$ | |
|---|---|---|
| | 1 min | 1 h |
| 0.0–15.3 | 1.00 | 1.00 |
| 15.3–30.5 | 1.18 | 1.23 |
| 30.5–46.0 | 1.31 | 1.40 |
| 46.0–61.0 | 1.40 | 1.52 |
| 61.0–76.0 | 1.47 | 1.62 |
| 76.0–91.5 | 1.53 | 1.71 |
| 91.5–106.5 | 1.58 | 1.78 |

### 3.3. Combining Environmental Loads

Design environmental loads for offshore fish farming installations can be derived from measured data at the actual deployment site, or design data specified by related classification or certification rules. To compose the design environmental loads, the dynamic loads must be realistically combined to include the maximum dynamic effect on each component of the fish farm.

Based on the BV rule for the purpose of load combination, the same return period acting in the same direction is generally to be assumed for combining the design environmental loads for the primary components of the fish farm [24]. On the other hand, the DNV rule introduces various combinations of environmental parameters (e.g., magnitude, return period, and direction) for wind, wave, and current that can be considered. Nevertheless, the most unfavourable combination of environmental loads shall be used for dimensioning the offshore fish farming installation [20]. A common practice to determine the extreme condition is by applying the worst combination of wind, wave, and current with approximately equal return period based on a statistical approach. However, caution should be exercised with a problem related to design criteria based on environmental statistics when the return period for the characteristic load effect is unknown for nonlinear dynamic systems. In general, the problem will lead to an inconsistent safety level for different design concepts and failure modes [20]. In this review paper, three major components: floating units, net and supporting system, and mooring system are discussed to establish applicable load combinations referred from the relevant rules and standards.

3.3.1. Load Combination for Designing Floating Units

In view of applicable classification rules for the load combination, the DNV rule for structural design of offshore units [20] may be used to combine dynamic environmental loads for floating units of offshore fish farms. The rule states that the combination of environmental loads can be obtained with available joint probability of environmental load components at the specified probability level that may be applicable to a specific site. Alternatively, a joint probability of environmental loads may be approximated by the combination of characteristic values for different load types. The load intensities for various types of loads can be combined according to the return period, or annual probability of exceedance, which is formulated simply as the inversion of the return period. For example, Table 6 shows the possible combinations of environmental loads with different return periods of wind, wave, current, and sea level for strength and accidental assessments based on the DNV rule associated with a minimum 20 year design life of the floating units.

**Table 6.** DNV's combinations of environmental loads for offshore units.

| Condition | | Combined Environmental Loads According to Return Period (Year) | | | |
|---|---|---|---|---|---|
| | | Wind | Waves | Current | Sea Level |
| Strength | A | 100 | 100 | 10 | 100 |
| | B | 10 | 10 | 100 | 100 |
| Accidental * | | ≥1 | ≥1 | ≥1 | ≥1 |

* Note that the DNV rule has its basis of immediate repair after damage occurred, and the damaged unit shall resist functional and environmental loads corresponding to a 1 year return period. Therefore, the accidental load combination for the damaged structure is determined as the most probable annual maximum value.

On the other hand, the ABS rule recommends load combinations depending on manned and unmanned floating fish farming installations, as shown in Table 7. As offshore fish farming installations must be designed for load scenarios and site-specific conditions encountered during transit, the design environmental conditions shall be defined as the extreme condition with a specific combination of wind, wave, and current for which the system is designed. The rule requires a minimum return period of 100 years to combine current, wind, and wave for manned floating fish farming installations, whereas a minimum return period of 50 years is required to combine current, wind, and wave for unmanned floating fish farms. In addition, if it is accepted by the coastal state, a minimum return period of 50 years can be specially considered for the manned floating fish farms.

**Table 7.** ABS's combinations of environmental loads for floating fish farms.

| Combination | Return Period (Year), Environmental Load | | |
|---|---|---|---|
| | Current | Wind | Wave |
| Manned floating fish farms | 100 | 100 | 100 |
| Unmanned floating fish farms | 50 | 50 | 50 |

3.3.2. Load Combination for Designing Net and Supporting System

In general, the maximum drag load on nets for offshore fish farms is, in most cases, caused by current and not by waves, while the floating units are in the opposite case. Therefore, an appropriate set of combinations, possibly different to the floating units, for dynamic loads shall be taken into consideration for designing of the net and its supporting systems. DNV's guidance for fish escape prevention from marine fish farms [15] states that the net and supporting system can be calculated and dimensioned in accordance with the load combinations presented in Table 8. For example, drag forces from water current flow must be checked for at least a minimum 50 year return period in combination with a 10 year return period of wind and wave loads, or the other way is a minimum 10 year return period of current in combination with a 100 year return period of wind and wave loads. Note that these combinations should be considered optional, as the net is outside the classification scope mentioned in Section 3.2.2.

**Table 8.** DNV's combinations of environmental loads for fish net pens.

| Combination | Return Period (Year), Environmental Load | | |
|---|---|---|---|
| | Current | Wind | Wave |
| 1 | Min. 50 | 10 | 10 |
| 2 | Min. 10 | 100 | 100 |

3.3.3. Load Combination for Designing Mooring System

Based on the technical standard for Scottish finfish aquaculture [29], two different combinations of environmental parameters can be considered, as given in Table 9, and the most unfavourable combination shall be used for dimensioning a mooring system.

**Table 9.** Combinations of environmental loads for mooring systems from Scottish finfish aquaculture.

| Combination | Return Period (Year), Environmental Load | |
|---|---|---|
| | Current | Wave |
| 1 | 50 | 10 |
| 2 | 10 | 50 |

Note that alternatively, the 50 year return period current shall be used when waves are not considered for designing the mooring system.

ABS's position mooring guide includes that the design environmental conditions for the mooring system shall be one of the following combinations that results in the most severe loading case:

- 100 year return period waves with associated wind and current.
- 100 year return period wind with associated waves and current.
- 100 year return period current with associated waves and wind.

Apart from the listed combinations, the rule states that additional design environmental load cases may be required to consider in areas with high current. In addition, a minimum return period of 50 years can be specially considered if it is accepted by the coastal state.

According to DNV's standard for offshore standard for mooring [23], environmental load combinations shall be considered in the mooring line response analysis for ultimate limit state (ULS) and accidental limit state (ALS), as given in Table 10. The most unfavourable combination of wind, wave, and current with a return period of 100 years or more shall be considered. Unfavourable conditions are those conditions that lead to higher mooring loads. As both the intensities and the directions of the environmental effects are significant, conservative conditions shall be applied if detailed information is not sufficient. The standard also specifies a combination used for Norwegian and UK sectors and some other extratropical locations, which is usually acceptable by employing both wind and waves with 100 year return periods, together with current with a 10 year return period. However, this combination may not be acceptable for sites with significant current loads.

**Table 10.** Combination of environmental loads considered for ULS and ALS from DNV.

| Combination | Return Period (Year), Environmental Load | | |
| --- | --- | --- | --- |
| | **Wind** | **Wave** | **Current** |
| ULS and ALS | Min. 100 | Min. 100 | Min. 100 |
| For Norway and UK sectors and some extratropical locations | 100 | 100 | 10 |

*3.4. Miscellaneous Load Conditions*

Floating structures for offshore fish farms may be needed to consider modes of conditions in preservice (loadout, transportation, installation) and inservice (inplace, maintenance, inspection) based on relevant rules for application [10]. For example, loads generated during the deployment process may include lifting components into the air, towing a system out to the site, and deploying the mooring system during pre-tensioning operations. In addition, accidental loads may occur through various situations, such as failure of the floatation elements, collision with service vessels or other vessels, failure of mooring lines and connector breakage, and impact loads such as slamming, sloshing [61,62] (for close containment tank), and breaking wave can be generated depending on fish pen types and site-specific conditions. Potential accidental loads may be necessary to consider for evaluation of the floating structures. Those preservice and inservice conditions can be investigated by using anticipated loads, including gravity loads together with relevant environmental loads. Moreover, effects of earthquake, thermal and ice loads, and marine growth loads may be considered as deemed necessary for the fish farming site.

In particular, marine growth may result in a significant load impact on floating structure members, nets, and mooring lines. The type and accumulation rate of marine growth at the design site can affect mass, weight, hydrodynamic characteristics (including shielding effect of nets), and drag coefficients. Therefore, marine growth is to be taken into consideration for components which are not subject to regular marine growth removal [12]. An appropriate load impact factor for marine growth should be included in the load analysis of long-term perspective. The thickness of the marine growth may be estimated in accordance with the specification for the actual location. If the data is not available, an alternative method to account for the marine growth may be used by increasing the weight of segments and increasing the drag coefficients, as exampled in DNV's offshore standard for mooring lines, as shown below [23].

- Mass of marine growth is determined by:

$$M_{growth} = \frac{\pi}{4}\left[\left(D_{nom} + 2\cdot\Delta T_{growth}\right)^2 - D_{nom}^2\right]\rho_{growth}\cdot\mu \tag{14}$$

- Submerged weight of marine growth is given by:

$$W_{growth} = M_{growth}\left[1 - \frac{\rho_s}{\rho_g}\right]g \tag{15}$$

where $\rho_g$ is the density of marine growth (may be set up 1325 kg/m$^3$), $\rho_s$ is the density of sea water, $g$ is the gravitational acceleration, $D_n$ is the nominal chain or wire rope diameter, $\Delta T_{growth}$ is the marine growth surface thickness, and $\mu$ = 2.0 for chain, and 1.0 for wire rope.

- Drag coefficient due to marine growth is estimated by:

$$C_{D\_growth} = C_D \left[ \frac{D_{nom} + 2 \cdot \Delta T_{growth}}{D_{nom}} \right] \tag{16}$$

The drag coefficient used in Equation (16) can be approximated with different types of chain and rope based on DNV's standard for mooring lines, as shown in Table 11.

**Table 11.** DNV's applicable drag coefficient ($C_D$) for chain and rope.

|  | Stud Chain | Studless Chain | Stranded Rope | Spiral Rope without Sheathing | Spiral Rope with Sheathing |
|---|---|---|---|---|---|
| $C_D$ | 2.6 | 2.4 | 1.8 | 1.6 | 1.2 |

Note that other methods for determining the increased drag coefficients due to marine growth may be accepted.

## 4. Global Performance Analysis Procedures and Methods

With offshore fish farms operating under extreme environmental conditions, it is necessary to conduct a global performance analysis and assessment of the entire offshore fish farming system to minimize structural failure and subsequent fish escape [34]. The analysis procedures may include quantifying hydrostatic stability analysis and hydrodynamic response analysis of the entire system, including moorings under frequency and time domains. In general, static loads, external hydrodynamic loads, internal dynamic loads (fluids stored onboard, ballast, major equipment items, etc.), and inertial loads of the structure are to be identified, and their magnitude and dynamic combinations are to be determined [9]. The suitability of the structure for all combinations of the dynamic loads must be evaluated using an acceptable analysis method, such as finite element analysis [34].

A global performance analysis of floating fish farming infrastructure is aimed to determine the overall effect of environmental loads on the entire system and its components, such as topside and floating units, mooring lines, and anchors. In general, the global performance analysis is to be carried out for all critical conditions, including preservice and inservice phases [10]. The following parameters shall be determined from the global performance analysis based on the ABS rule for floating fish farming installation [10]:

1. Motions of the floating fish farming installation in six degrees of freedom.
2. Mooring line tensions, including the maximum and minimum tensions and fatigue loads for mooring component design.
3. Critical global forces and moments, or equivalent design wave heights and periods as appropriate for the hull structural analysis.
4. Hull hydrodynamic pressure loads for global structural analysis.
5. Accelerations for the determination of inertia loads.

The hydrodynamic loads used in the global performance analysis may be obtained through:

1. Hydrodynamic analysis for large bodies based on radiation/diffraction theory using panel models.
2. The Morison equation for slender members, external hull appurtenances, and viscous hull drag with well documented drag coefficients $C_d$ and inertia coefficients $C_m$.
3. Computational fluid dynamics (CFD) or model test to determine hydrodynamic loads and coefficients on some innovative or unconventional structural components.

Either frequency or time domain methods, or a combination of both, may be used in the global performance analysis. However, for those cases that have highly nonlinear effects,

a time domain analysis is normally required. Figure 2 shows an example of the global performance analysis procedure that can be used for designing offshore fish pens [37].

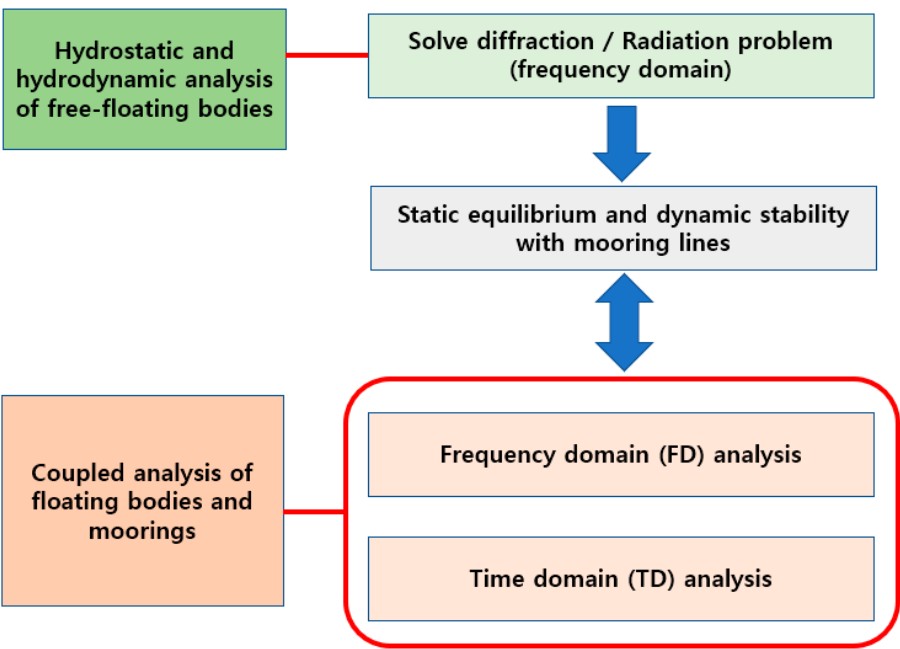

**Figure 2.** Example of global performance analysis procedure.

*4.1. Hydrostatic Analysis*

Force balance between structure weight and buoyancy (i.e., hydrostatic equivalence) is the starting point in designing any floating structure for offshore use [22]. Therefore, the hydrostatic analysis for offshore fish farming installations aims to find the force balanced condition for fish containment systems in either surface or submerged configurations, depending on their operating modes. Usually, this effort is trivial, but important for the success of subsequent performance analyses [22].

According to the technical guidance for offshore aquaculture installations written by Fredrisson and Beck-Stimpert [34], the hydrostatic analysis shall consider weight, flotation, and reserve buoyancy, and different approaches have to be taken depending on the selection of the floating system. For example, if the system is placed at the surface with substantial displacement characteristics, the appropriate transverse and longitudinal metacentric heights must be positive. When the system is deployed in the underwater configuration, the transverse and longitudinal (if not symmetric) centre of buoyancy must be positioned higher than the transverse or longitudinal centre of gravity. If tension leg moorings are incorporated into the system design, an appropriate hydrostatic analysis must be performed with specification of mooring leg configuration. If flotation buoys are used at the water surface, the waterline and reserve buoyancy shall consider the flotation buoys in the hydrostatic calculation. For the hydrostatic equilibrium calculation, the influence from mooring pre-tensions is also needed to include as a part of load balance.

In general, hydrostatic parameters given in the below list shall be obtained during the hydrostatic analysis, as each of the parameters has a vital role in the stability of the free-floating condition.

- Vertical, longitudinal, and transverse centre of gravity.
- Vertical, longitudinal, and transverse centre of buoyancy.
- Mass displacement.
- Volume displacement.
- Waterplane area and metacentric radius.
- Metacentric height.

*4.2. Hydrodynamic Analysis*

Many fish farming systems have flexible components, such as net, chain, rope, and even floaters (if a flexible material is used), due to their resilient features against dynamic forces. Furthermore, in the offshore environment, waves and currents can generate severe dynamic forces that cause large displacements and rotations as the system components exhibit movement relative to the fluid velocities and accelerations. These dynamic effects make it difficult to obtain accurate global force distributions through analytical procedures. Therefore, an appropriate analysis technique shall be employed to assess the hydrodynamic responses [34,63].

Many of the existing technical guidelines for offshore floating installations require a hydrodynamic analysis in the time domain and frequency domain [9,22]. If the fish farming system consists of a large volume structure, the wetted surface geometry can be treated as a three-dimensional boundary panel element model, called a panel model. The linear potential wave theory (see Section 3.2.1) and numerical analysis programs (e.g., Ansys AQWA [44], AquaSim [64], WAMIT [65], SIMO [66]) are often employed to solve the hydrodynamic radiation and diffraction problems for the interaction of surface waves with floating structures. In addition, the panel model may include Morison elements for the slender components (i.e., diameter smaller than one-fifth wavelength) of submerged components, as long as the actual location and dimensions are implemented.

4.2.1. Frequency Domain Analysis

If nonlinear effects can be considered insignificant, or if such loads can be satisfactorily described in a linear analysis, a frequency domain analysis may be acceptable. In the frequency domain analysis, transfer functions for structural responses are established with a sufficient number of wave directions in radial spacing and wave periods. According to the recommended practice from DNV [22], the selection of wave periods for the frequency analysis is usually performed with a basis in:

- Peak period in wave spectrum.
- Location of natural periods.
- Geometrical considerations (diameters of columns, spacing between columns, wave headings, etc.).

In general, the frequency domain analysis shall cover six degrees of freedom of the floating fish farming installation, in both the wave frequency and the low frequency domains. The linear wave theory is commonly used in the wave frequency analysis to evaluate the wave frequency responses of floating fish farming installations [37]. The low frequency motion analysis can be performed to evaluate the effects caused by wind dynamics and wave drift forces [9]. Hydrodynamic loads on large bodies are usually obtained in the frequency domain based on radiation/diffraction theory using a panel model. In general, unless the standards and guidelines specify a panel size, a convergence study should be performed to find an appropriate panel size (i.e., mesh size) to obtain accurate results with minimal use of computing resources.

For hydrodynamic loads on slender bodies, and viscous drag loads on the hull structures and fish net pens, the Morison equation can be used. However, the viscous drag loads used in the frequency domain analysis should be linearized [9,37]. In general, the viscous drag load component is caused by viscosity and the relative velocity between fluid particles and the surface of structure. Thus, the viscous drag force applied to the unit length of Morison-type structures (i.e., applicable Morison equation) can be expressed as:

$$dF_D = \frac{1}{2}C_D\rho D\left|V_f - V_s\right|\left(V_f - V_s\right) \tag{17}$$

in which $C_D$ is the drag coefficient, $\rho$ is the fluid density, $D$ is the structure diameter, $V_f - V_s$ is the transverse relative velocity between fluid velocity $V_f$ and structure velocity $V_s$. In order to consider a reasonable viscous damping effect for the Morison-type structures, an

appropriate viscous drag coefficient $C_D$ can be obtained from model tests, or computational fluid dynamics (CFD) analyses.

In the frequency domain, the nonlinear term of $\left|V_f - V_s\right|\left(V_f - V_s\right)$ should be linearized, and one may adopt the following format [42]:

$$dF_D = \frac{1}{2}C_D\rho D\alpha V_{rms}V_f - \frac{1}{2}C_D\rho D\alpha V_{rms}V_s \tag{18}$$

where $\alpha$ is the factor expressed as $\sqrt{8/\pi}$ [67], $V_{rms}$ is the root mean square of the transverse relative velocity, and can be calculated iteratively based on the wave spectrum and the response amplitude operators (RAOs) of the structural transverse velocity [42]. The first term on the right-hand side of Equation (18) can be considered as the external fluid force, while the second item can be converted to the linear fluid damping.

The frequency domain dynamic analysis involves a linear combination of mean, low frequency, and wave frequency components of force and response. Throughout the frequency domain analysis, transfer functions for frequency dependent-excitation forces (first and second order), added mass, and damping (potential and viscous) can be generated. The analysis results can be in the form of motion response amplitude operators (RAOs) [68,69]. In a frequency domain analysis, the governing equation for the motion of a floating body in six degrees of freedom can be given by:

$$\left[-\omega^2(M + A(\omega)) + i\omega B(\omega) + C\right]\zeta(\omega) = F(\omega) \tag{19}$$

where $M$ is the $6 \times 6$ structure mass matrix, $\omega$ is the angular frequency, $A(\omega)$ is the frequency-dependent $6 \times 6$ hydrodynamic added mass matrix, $B(\omega)$ is the frequency-dependent $6 \times 6$ damping matrix, $C$ is the $6 \times 6$ hydrostatic stiffness matrix, $\zeta(\omega)$ is the dynamic response vector, and $F(\omega)$ is the dynamic load vector in six degrees of freedom.

A series of linear algebraic equations based on Equation (19) can be solved to obtain the harmonic response of the floating body under incoming regular waves. Then, RAOs, which are normalized to the wave of unit amplitude, can be calculated by solving the complex matrix equations in six degrees of freedom, determined by:

$$\text{RAOs}(\omega) = F(\omega)\left\{-\omega^2(M + A(\omega)) + i\omega B(\omega) + C\right\}^{-1} \tag{20}$$

The outputs of the traditional frequency domain analysis are typically excitation forces/moments, added mass/moments and potential damping, and motion RAOs. If a hybrid (inclusive Morison elements) model has been made, the Morison loads can be added directly into the results. This means that the model can include linearised finite wave amplitude effects and viscous damping contributions.

The frequency domain analysis, however, usually does not include nonlinear interactions, since the processes are decoupled. It has been shown that the nonlinear wave-current interaction effects on fish containment structures can represent a significant loading component when applied to such systems [37]. Therefore, when frequency domain analyses are applied to evaluate dynamic loads, design factors should consider nonlinear interactions and other uncertainties [34].

### 4.2.2. Time Domain Analysis

A time domain analysis is the preferable approach to include nonlinear effects in the global performance analysis for floating fish farming installations. These nonlinear effects include drag forces of floating bodies, drag of fish nets, nonlinear restoring forces of mooring lines, effects of motion suppression devices or components (e.g., heave plates), and coupling effects of the floating body and mooring system. When strong nonlinear responses are expected within the floating fish farms, a time domain analysis must be performed [9]. The advantage of the time domain analysis is that it can capture higher order

load effects (e.g., second order drift) and gives the response statistics without assuming the response distribution.

In the time domain analysis, the associated wave spectrum is to be transferred to random time series to simulate irregular wave heights and kinematics. The maximum responses should be predicted by an appropriate distribution curve fitted to the simulation results, or other recognized statistical techniques. In cases where the time domain analysis is time-consuming, only critical events can be simulated by a refined model with a short time observation [21,37]. However, the time domain analysis should be performed over a period sufficient to achieve stationary statistics, especially for low frequency responses. Multiple realizations of the same conditions may be required to generate adequate data for a statistical analysis and to ensure the simulation consistency. Thereby, a designer shall demonstrate the adequacy of the selected simulation time duration and the number of realizations [9].

Normally, irregular sea-states in offshore conditions are stationary in time over a duration of 3 h in the full scale condition [21,22]. Thereby, many time domain analyses for offshore fish farms observed 3 h responses of the irregular waves so that they can represent stationary sea conditions [6,36,68,70]. In order to demonstrate a random time series of the irregular sea-states, an appropriate wave spectral shape can be estimated from datasets, whichever is available at the target farm site. If the datasets are not available, parameterized analytic formula (frequently applied for wind seas), such as the Pierson–Moskowitz (PM) spectrum and the JONSWAP spectrum, can be adopted with consideration of the geographical area, local bathymetry, and severity of the sea-state [71].

In order to specify the strength specification of offshore fish farms, the extreme conditions in the long-term variation between 10 and 100 years return periods are typically considered in developing design conditions [34]. The long-term variation can be described in terms of scatter diagrams from the governing sea-state parameters, such as significant wave height and zero-crossing period ($H_s$ $T_z$). A probability distribution method, such as Weibull or Gumbel, is typically applied for the distribution of significant wave height in the long-term prediction [21,68]. For example, the annual extremes of significant wave height can be assumed to follow a Gumbel distribution, and the probability distribution function (PDF) of the Gumbel distribution can be written with respect to $H_s$ as:

$$PDF(H_s) = \exp\left\{-\exp\left[-\frac{(H_s - U)}{A}\right]\right\} \tag{21}$$

where $A = \sigma/1.283$, $U = \mu - 0.557A$, $\sigma$ is the standard deviation, and $\mu$ is the mean value.

The zero-crossing period $T_z$ corresponding to the significant wave height $H_s$ can be defined by a joint probability density function $f_{H_s,T_z}(H_s, T_z)$ of the significant wave height and zero-crossing period [72], that can be written as:

$$f_{H_s,T_z}(H_s, T_z) = f_{H_s}(H_s).f_{T_Z|H_s}(T_z|H_s) \tag{22}$$

where $f_{H_s}(H_s)$ is the marginal probability distribution of the significant wave height $H_s$, and $f_{T_Z|H_s}(T_z|H_s)$ is the probability distribution of the spectral zero-crossing period $T_z$ conditional on the significant wave height $H_s$. The distribution of the zero-crossing period $T_z$ for any sea-state defined by $H_s$ is given by:

$$f_{T_Z|H_s}(T_z|H_s) = \frac{1}{\sigma T_z \sqrt{2\pi}} exp\left\{-\frac{[\ln T_z - \mu]^2}{2\sigma^2}\right\} \tag{23}$$

where the parameters of mean value $\mu = \mu(H_s)$ and standard deviation $\sigma = \sigma(H_s)$, defined for each $H_s$ in the log-normal distribution, can be estimated by fitting techniques [72] from any available sea-state dataset, as:

$$\mu(H_s) = a_1 + a_2.h_s{}^{a_3} \tag{24}$$

$$\sigma(H_s) = b_1 + b_2.exp(b_3.h_s) \tag{25}$$

where $a_1$, $a_2$, $a_3$ and $b_1$, $b_2$, $b_3$ are estimated coefficients from the raw data by the nonlinear least-squares method [73].

Note that the extreme value estimation should be compared with results from alternative methods, including laboratory tests. It can be a practical challenge, but in principle, it is possible to generate nonstationary conditions in laboratory conditions. Therefore, caution should be taken to properly interpret the response statistics from such tests. Extreme values in a random process are random variables with a statistical variability. Therefore, a sample extreme from the 3 h simulation proposed here must be interpreted as just one sample out of many [21].

*4.3. Mooring System Analysis*

The mooring system affects the stresses acting on the structural members and the behaviour of fish farming installations during adverse weather conditions that can affect production, profitability, and the safety of workers [74]. Therefore, the mooring system is an important part of fish farming installations and should be carefully designed. Floaters, net, and mooring components should be designed and mechanically linked together. Their dynamic responses cannot be considered individually, as every component influences each other. Therefore, the mooring design shall be "site specific", and carefully combined with the choice of floater and net types with respect to the survivability of fish stocks in major storms at exposed sites [75]. A better understanding of dynamic responses and more sophisticated analyses of the mooring system will largely reduce the risks associated with offshore fish farming [75].

Mooring systems can be classified either as a single-point mooring system or as a spread mooring system [76]. The former is used mainly for ship-shaped offshore fish farming installations, while the latter is commonly used for semisubmersibles or an array of fish pens where the environmental loadings acting on the floating units are relatively insensitive to environment load directions. Regardless of mooring types, the mooring system usually consists of partially grounded lines on the seabed with their bottom-end fixed to the seabed by means of anchors, and their top-end connected to fairleads at the floater. The common characteristics of these mooring lines are that they are totally flexible and highly nonlinear in geometry, thus, special methods are required for their analysis.

A mooring system can be designed either by a quasi-static mooring analysis, or a full dynamic response analysis. The choice of analysis is highly dependent on the range of errors from the techniques, and the amount and accuracy of input data. In general, a well analysed quasi-static analysis will be much better than a dynamic analysis performed with limited data [77,78]. A designer should determine the extreme offset and line tension in a consistent manner with the chosen analysis method. According to DNV's recommendation, the quasi-static analysis is generally required to determine the mooring line responses corresponding to mean and low-frequency platform displacements, while the dynamic mooring analysis is usually required to determine mooring line responses corresponding to wave-frequency displacements of the platform [23]. Some commercial software packages, such as FhSim (https://fhsim.no/, accessed on 11 November 2022), Orcaflex [79], AquaSim (https://aquasim.no/, accessed on 10 February 2023), and Ansys AQWA [44], can be used for analysing the mooring system of a fish farming structure.

4.3.1. Quasi-Static Method

When a quasi-static analysis method is used, the tension in each mooring line is to be calculated with respect to the maximum excursion from each design condition. In this approach, the tension force in the mooring line is statically calculated, corresponding to the offset of floating structures moved by wave-induced motions. In general, dynamic actions on the mooring line associated with mass, damping, and fluid acceleration are not considered in the quasi-static analysis, and the anchor position is assumed to be fixed at the seabed in the mooring analysis. However, various conditions, such as the type of floating

structures, mooring systems and line configuration, and water depth are to be considered when the quasi-static method is selected to predict mooring loads and the effect on the mooring system [12].

According to the DNV's standard for mooring systems [23], the quasi-static mooring analysis must take account of:

- The displacement of the upper end (i.e., fairlead) point of the mooring line due to the floating unit's motions.
- The weight and buoyancy of the mooring line components.
- The elasticity of the mooring line components.
- Reaction and friction forces from the seabed.

The standard also states that the statistics of the mooring response through the quasi-static method can be expressed as the line tension $T$ at a point in the line from a function of $X$, where $X$ is the horizontal distance between the lower (i.e., anchor) and upper (i.e., fairlead) points. It can be expressed as $T_{QS}[X]$. In addition, two components of the line tension can be considered:

(1) $T_{C-mean}$: The mean line tension due to pre-tension, and mean environmental loads caused by static wind, current, and mean wave drift forces.

(2) $T_{C-dyn}$: The dynamic line tension induced by low-frequency and wave-frequency motions.

Thereby, the dynamic tension $T_{C-dyn}$ within the quasi-static mooring line analysis can be expressed as:

$$T_{C-dyn} = T_{QS}[X] - T_{C-mean} \tag{26}$$

The relationship between horizontal offset $X$ and the corresponding tension components are illustrated in Figure 3.

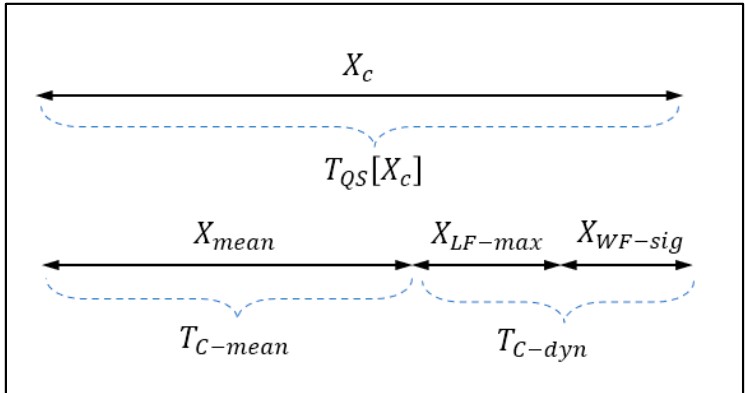

**Figure 3.** Relationship between horizontal distance $X$ and the corresponding tension components in quasi-static analysis [23].

The following response statistics are to be determined in each environmental state considered:

(1) $X_{mean}$: The mean horizontal offset of the upper end point of the mooring line from the anchor.

(2) $\sigma_{X-LF}$: The standard deviation of horizontal, low-frequency motion of the upper end point in the mean mooring line direction.

(3) $\sigma_{X-WF}$: The standard deviation of horizontal, wave-frequency motion of the upper end point in the mean mooring line direction.

A Gaussian process model can be applied in the development of the line tension from the aforementioned response statistics. On this basis, the significant and maximum low-frequency excursion can be defined as:

$$X_{LF-sig} = 2\sigma_{X-LF}, \quad X_{LF-max} = \sigma_{X-LF} \cdot \sqrt{2 \ln N_{LF}} \tag{27}$$

where $N_{LF}$ is the number of low-frequency oscillations during the duration of the environmental state, which is normally taken as 3 h.

The significant and maximum wave-frequency excursion can be defined in a similar manner, as:

$$X_{WF-sig} = 2\sigma_{X-WF}, \quad X_{WF-max} = \sigma_{X-WF} \cdot \sqrt{2 \ln N_{WF}} \tag{28}$$

where $N_{WF}$ is the number of wave-frequency oscillations during the duration of the environmental state.

The offset $X_C$ can be expresses as the larger of $X_{C1}$ and $X_{C2}$:

$$X_{C1} = X_{mean} + X_{LF-max} + X_{WF-sig} \tag{29}$$

$$X_{C2} = X_{mean} + X_{LF-sig} + X_{WF-max} \tag{30}$$

The motion amplitude is highly dependent on the stiffness and damping of the mooring system. Thereby, the mooring system has to be properly modelled in the analysis [12,23]. The general steps in the quasi-static mooring analysis are as follows [75]:

(a)  Define the mooring geometry and mooring excursion/force equations.
(b)  Apply the mean environmental force to the system and calculate the excursion (offset).
(c)  Apply the periodic wave forces and response amplitude to the system.
(d)  Calculate the line tensions resulting from this maximum excursion.
(e)  Compare the line tensions with the minimum breaking load of the riser components.
(f)  Calculate the maximum peak anchor loads for each riser and direction.
(g)  Introduce a safety factor (generally 2.0) when calculating the line strengths.
(h)  Recalculate the maximum peak line loads with one line broken, or after a line failure.
(i)  If the proposed mooring specification fails the safety factor test, then try a new specification.

### 4.3.2. Dynamic Method

The dynamic method can be used to analyse mooring line responses. According to the DNV's standard for mooring systems [23], the dynamic mooring analysis must take account of those mentioned in the quasi-static method, as well as additional consideration of:

- Hydrodynamic drag forces acting on the mooring line components.
- Inertia forces acting on the mooring line components, including any buoyancy elements.
- On the other hand, the BV rule for fish farms [24] indicates that a dynamic approach can account for all relevant time varying and nonlinear effects, including:
- Time varying effects of the wave and wind exciting force (wave and wind spectra).
- Damping effects on the fish farm.
- Nonlinear restoring forces of the anchoring lines.

For the dynamic approach, time domain simulations are generally used, but frequency domain or combined time and frequency domain approaches may be acceptable [12]. When the time domain simulations are carried out, a time interval is to be chosen as small enough to cover first order wave effects [24]. In addition to this, the BV rule for fish farms states that a dynamic mooring simulation is to be extended over the total time duration of the sea-state; in any case, it is not to be less than 1 h, excluding the transient phase. A statistical analysis of the results (line tensions, surge, sway, and yaw of the fish farm) is to be performed in order to derive the most probable largest values of the parameters over the sea-state duration.

Based on the DNV's standard [23], when the dynamic mooring analysis is applied, the dynamic tension force $T_{C-dyn}$ can be further defined as:

$$T_{C-dyn} = T_{QS}[X_C - X_{WF-max}] - T_{C-mean} + T_{WF-max} \tag{31}$$

where $T_{QS}[X_C - X_{WF-max}]$ is the quasi-static tension force calculated with the upper end position with allowance of the maximum wave-frequency excursion, and $T_{WF-max}$ is the maximum wave-frequency tension force. Note that the term of maximum wave frequency tension force is added in the equation to provide a remainder of the standard deviation of

wave frequency tension force in the function of the excursion about the wave frequency motion taking place. The maximum wave frequency tension $T_{WF-max}$ can be expressed as:

$$T_{WF-max} = \sigma_{T-WF}[X_C - X_{WF-max}] \sqrt{2 \ln N_{WF}} \tag{32}$$

Figure 4 shows the relationship between horizontal offset $X$ and corresponding dynamic tension components in the dynamic method [23].

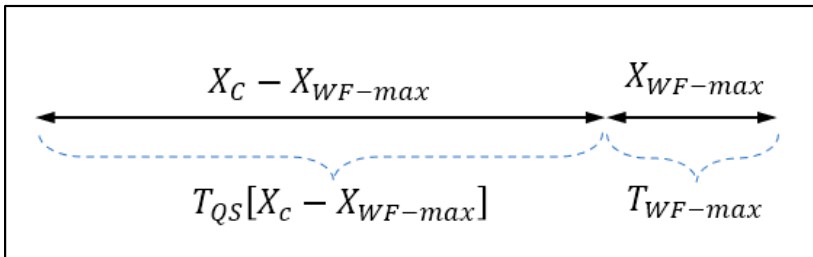

**Figure 4.** Relationship between horizontal distance X and the corresponding dynamic tension components in dynamic analysis.

It should be noted that the mean tension $T_{C-mean}$ from the frequency domain analysis and the time domain analysis may be different if the mooring dynamics in the time series are nonsymmetrical.

4.3.3. Mooring Assessment

In general, the design process of a mooring system inevitably involves many iterations [50]. An initial design of mooring layout and configuration can be proposed based on the previous experiences, or a preliminary mooring analysis of the coupled system (including floating units, net, and mooring lines) can be conducted with several load cases to cover the extreme scenarios. Although the worst case can be screened from the preliminary analysis, the maximum mooring line tension shall be multiplied by a certain partial safety factor, so that the result can be compared against the maximum breaking load as a safety check of the ultimate limit state (ULS) [12,23]. If the ultimate load of the mooring system is acceptable, the accidental limit state (ALS) of mooring lines should also be checked. Checking the ALS implies that the offshore floating installation shall have adequate safety redundancy upon the possible loss of mooring lines [50].

According to the DNV's standard for mooring systems [23], two consequence classes are introduced for the ULS and ALS assessments:

- Class 1: where mooring system failure is unlikely to lead to unacceptable consequences, such as loss of life, collision with an adjacent platform, uncontrolled outflow of oil or gas, capsize or sinking (assumed no fish in the pen when the pen is dormant and being cleaned at a service draft).
- Class 2: where mooring system failure may well lead to unacceptable consequences (assumed fish onboard when the pen is under a normal operation).

Different partial safety factors are introduced based on the consequence classes and types of analysis, as presented in Table 12.

The standard also indicates that instead of the separate static and dynamic safety factors given in Table 12, a common value of 1.3 should be applied if the mean tension exceeds two-thirds of the dynamic tension, and applying a dynamic analysis in the consequence class one. Moreover, a single-point mooring system is designed without redundancy and, consequently, ALS is not an applicable design condition. The system may be accepted by further multiplying a factor of 1.2 to the safety factors given in Table 12, and the loss of the mooring system should not result in a major pollution or damage to the floating unit. Emergency disconnection systems for moorings shall be required. Further, the main propulsion of the unit shall be in operation.

**Table 12.** DNV's partial safety factor for ULS and ALS assessment of mooring system.

| Assessment | Consequence Class | Type of Analysis | Partial Safety Factor on Mean Tension ($\gamma_{mean}$) | Partial Safety Factor on Dynamic Tension ($\gamma_{dyn}$) |
|---|---|---|---|---|
| ULS | 1 | Dynamic | 1.10 | 1.50 |
| | 2 | Dynamic | 1.40 | 2.10 |
| | 1 | Quasi-static | 1.70 | |
| | 2 | Quasi-static | 2.50 | |
| ALS | 1 | Dynamic | 1.00 | 1.10 |
| | 2 | Dynamic | 1.00 | 1.25 |
| | 1 | Quasi-static | 1.10 | |
| | 2 | Quasi-static | 1.35 | |

ABS's standard for mooring systems [12] states similar safety factors for the offshore floating production installations, as shown in Table 13. However, ABS can be differentiated from DNV, as it requires different safety factors depending on design conditions: all intact, one broken line at the new equilibrium position, or one broken line in transition, rather than the corresponding tension components. Nevertheless, both DNV and ABS standards require higher safety factors when the quasi-static analysis is employed to design the mooring system, when compared to the dynamic analysis.

**Table 13.** ABS's safety factor of mooring lines for offshore floating production installations.

| Design Condition | Type of Analysis | |
|---|---|---|
| | Quasi-Static | Dynamic |
| All intact (ULS) | 2.7 | 2.25 |
| One broken line at new equilibrium position (ALS) | 1.8 | 1.57 |
| One broken line in transition (ALS) | 1.4 | 1.22 |

It is worth noting that fish farming installations use synthetic fibre ropes for interconnection between fish pens, as well as for moorings. In general, synthetic materials have time-varying properties, and thus their hydrodynamic behaviours tend to be highly nonlinear. Thus, DNV's offshore standard for mooring systems [23] indicates that it is the synthetic rope manufacturer's responsibility to take a sufficient number of samples of the synthetic fibre rope for testing to establish the synthetic fibre rope properties. Requirements regarding testing are given in DNV-OS-E303 [80]. The change in length testing of the synthetic fibre rope should be defined based on the actual requirements of the mooring design, so that accurate ULS and ALS may be determined. ABS's guidance notes on the application of synthetic fibre rope for offshore mooring [81] also requires a similar testing scope as that of DNV's.

**5. Concluding Remarks**

This paper reviews existing design guidelines and technical guidance from maritime classification rules/standards and national and international standards which can be used for a design basis of offshore fish farming installations. It primarily focuses on the maritime classification rules/standards for offshore fish farming, and supplementally reviews national and international finfish farming standards, since national standards shall be considered as a volunteering application depending on the jurisdiction of governing localities for offshore fish farms.

In view of an offshore engineering perspective, this review paper compiles standards and guidelines primarily focusing on design criteria and global performance analysis procedure and methods for reasoning and decisions employed to develop offshore fish

farming infrastructure. This paper also includes limitations and cautions for the use of these references.

There are inconsistencies in different design guidelines, particularly in the way of environmental load combinations and partial safety factors for mooring assessment. More efforts are needed to provide a unified guideline, as well as a more rational design approach, through direct analyses in designing offshore fish pens in order to gain industry confidence with regard to offshore farms. Moreover, it is noted that this paper did not include design criteria such as fatigue and risk assessments, which are highly dependent on the selection of construction materials and fish farming methods. Future studies will review more detailed design criteria along with material choices and operating methods.

By providing comprehensive information on the existing design standards and guidelines, as well as commentary on their applicability, this paper should be useful to aquaculture engineers and designers when developing offshore fish farming infrastructure. While these standards and guidelines shall be considered as a volunteering application depending on the jurisdiction of governing localities for offshore fish farms, rationally supported and scientifically evidenced technical guidance will help to gain industry confidence in offshore fish farming.

**Author Contributions:** Conceptualization, Y.-I.C. and C.-M.W.; methodology, C.-M.W.; software, Y.-I.C.; validation, H.Z., N.A., H.K., D.-S.J. and P.A.A.; formal analysis, Y.-I.C.; investigation, Y.-I.C.; resources, P.A.A.; data curation, Y.-I.C. and H.K.; writing—original draft preparation, Y.-I.C. and C.-M.W.; writing—review and editing, H.Z., N.A., H.K., D.-S.J. and P.A.A.; visualization, H.K. and Y.-I.C.; supervision, J.B.; project administration, Y.-I.C.; funding acquisition, J.B. and C.-M.W. All authors have read and agreed to the published version of the manuscript.

**Funding:** This research was funded by the Blue Economy Cooperative Research Centre of Australia, grant number CRC-20180101.

**Institutional Review Board Statement:** Not applicable.

**Informed Consent Statement:** Not applicable.

**Data Availability Statement:** Not applicable.

**Acknowledgments:** The authors acknowledge the financial support of the Blue Economy Cooperative Research Centre, established and supported under the Australian Government's Cooperative Research Centres Program, grant number CRC-20180101.

**Conflicts of Interest:** The authors declare no conflict of interest.

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
