# Peer review of "Offshore Fish Farms: A Review of Standards and Guidelines for Design and Analysis"

_jmse, doi:10.3390/jmse11040762_

Round 1

Reviewer 1 Report

The authors have attempted to summarize guidelines for design and analysis of aquaculture structures in the open ocean. The paper is well laid out and straightforward to read and understand.  There are formatting errors (such as references not found) and some typographical errors (Such as use of filed instead of field).

The authors have used "guidelines" and "rules" interchangeably, which is not correct. Conformance to guidelines is not mandatory, as opposed to rules. Also note that many stationary structures do not need to conform to rules. Design following Class stipulated guidelines certainly will help with insurance, financing and during inspection.

The environmental force formulation section is missing a key discussion of forces on netted structures. There is ongoing discussion in the literature about using Morison's equation for individual strands vs. using a mean drag coefficient for a net panel. The authors should add a section focusing on forces on porous netted structures. I note that they have done some coverage of this topic when talking about current forces but it is not comprehensive. Also, how reliable is Eq. 11 (Cd for netted panels)?  Has this been endorsed by class guidelines?

Use of Newman's approximation is misrepresented in the text. Newman's method was developed originally for very small difference frequency values close to resonance frequency in surge and sway.  This hence assumes that the QTF is a banded matrix, and the approximation thus applies to off-diagonal terms that are very close to the diagonals. Two related aspects are whether the approximation is applicable to netted structures, and to structures in shallow water. Authors should consider these when discussing this approximation.

When discussing wind and current loads, authors should add some information on what time-averaged mean values are recommended.

Also, authors should comment on recommendations on calculating windage (area exposed to wind), as they can vary from one class society to another.

When discussing mooring analysis, authors should talk about methods that are suited for synthetic materials, as many installations will have synthetic ropes for interconnection as well as for mooring.  Synthetic materials suffer from time varying properties and their stiffness behavior can be nonlinear.

Author Response

Offshore Fish Farms: A review of Standards and Guidelines for Design and Analysis

Y.I Chu, C.M. Wang, H. Zhang, N. Abudussamie, H. Karampour, D.S. Jeng, J. Baumeister, P.A. Aland

The authors wish to thank the editor and the reviewers for their insightful comments and suggestions on our paper. We have revised the paper to address all these comments and suggestions. Below are our point-by-point replies to the reviewers’ comments.

Reviewer #1

(1) The authors have attempted to summarize guidelines for design and analysis of aquaculture structures in the open ocean. The paper is well laid out and straightforward to read and understand.  There are formatting errors (such as references not found) and some typographical errors (Such as use of filed instead of field).

Authors reply: Formatting errors and typo errors have been corrected.

(2) The authors have used "guidelines" and "rules" interchangeably, which is not correct. Conformance to guidelines is not mandatory, as opposed to rules. Also note that many stationary structures do not need to conform to rules. Design following Class stipulated guidelines certainly will help with insurance, financing and during inspection.

Authors reply: Authors agree with the reviewer’s point that rules and guidelines are different. However, the authors would like to point out that maritime classification rules are considered optional in the offshore fish farming industry. Although it is optional, class service entries require strict rule compliance for design approval and product certification, which will give a great benefit for gaining credit from clients and commercializing product. Section 2.1 mentioned this benefit and includes more argument as: “Note that the entry of classification services for offshore fish farming infrastructure is considered optional. However, in order to obtain the design approval and product certification, relevant classification rules shall be strictly complied. In this review paper the term “rule” will be used when referring to classification rules for the design guidance.”

(3) The environmental force formulation section is missing a key discussion of forces on netted structures. There is ongoing discussion in the literature about using Morison's equation for individual strands vs. using a mean drag coefficient for a net panel. The authors should add a section focusing on forces on porous netted structures. I note that they have done some coverage of this topic when talking about current forces but it is not comprehensive. Also, how reliable is Eq. 11 (Cd for netted panels)?  Has this been endorsed by class guidelines?

Authors reply: According to the classification rules, cage net is beyond classification scope (see Sec. 2.1). Therefore, this review paper on standards and guidelines does not address deeply on methods to estimate forces on nets. Instead, sections 3.2.2 and 3.3.2 partially mentioned about simplified methods to approximate current velocity reduction, drag coefficient, and load combination factors for net and its supporting structure.

With respect the reviewer’s comments, the revision includes some references to introduce other methods such as the Morison equation and screen type method in section 3.2.2. with a comment on maritime classification perspective: “Note that the given Eq. (16) is an empirical approach that can be highly dependent on the experimental setup. Other methods can also be adopted such as by utilizing the Morison equation to mesh line/bar elements [57] or screen type method by applying load to membrane net panels [58]. However, to the authors’ knowledge, such approaches are not endorsed by maritime classifications and net is considered outside the classification scope.”

[57]     Tsukrov, I.; Eroshkin, O.; Fredriksson, D.; Swift, M.R.; Celikkol, B. Finite Element Modeling of Net Panels Using a Consistent Net Element. Ocean Eng. 2002, 30, 251–270, doi:10.1016/S0029-8018(02)00021-5.

[58]     Kristiansen, T.; Faltinsen, O.M. Modelling of Current Loads on Aquaculture Net Cages. J. Fluids Struct. 2012, 34, 218–235, doi:10.1016/j.jfluidstructs.2012.04.001.

(4) Use of Newman's approximation is misrepresented in the text. Newman's method was developed originally for very small difference frequency values close to resonance frequency in surge and sway.  This hence assumes that the QTF is a banded matrix, and the approximation thus applies to off-diagonal terms that are very close to the diagonals. Two related aspects are whether the approximation is applicable to netted structures, and to structures in shallow water. Authors should consider these when discussing this approximation.

Authors’ reply: The relevant sentence is revised to reflect the reviewer’s comment. It now reads as: “In particular, for netted structures with new floater concepts and/or relatively shallow water installations, caution should be taken as to whether Newman’s approximation is applicable.”

(5) When discussing wind and current loads, authors should add some information on what time-averaged mean values are recommended. Also, authors should comment on recommendations on calculating windage (area exposed to wind), as they can vary from one class society to another.

Authors’ reply: The revision provides DNV and ABS’s recommendations on average times to specify wind speed and a simple equation to calculate wind pressure based on shape and height coefficients of windage given in Table 4 and Table 5.

(6) When discussing mooring analysis, authors should talk about methods that are suited for synthetic materials, as many installations will have synthetic ropes for interconnection as well as for mooring.  Synthetic materials suffer from time varying properties and their stiffness behavior can be nonlinear.

Authors’ reply: Authors agree that many fish farming installations use synthetic fibre ropes for fish pen interconnection as well as mooring lines. Since synthetic material has time varying properties, their hydrodynamic behaviours tend to be highly nonlinear. Therefore, DNV and ABS classification guidelines indicate that it is the manufacturer’s responsibility to provide nonlinear properties obtained from physical experimental tests as required by classification. The revision includes this argument in section 4.3.3 as: 

“It is worth noting that fish farming installations use synthetic fibre ropes for interconnection between fish pens as well as for moorings. In general, synthetic materials have time varying properties and thus their hydrodynamic behaviours tend to be highly nonlinear. Thus, DNV’s offshore standard for mooring system [24] indicates that it is the synthetic rope manufacturer’s responsibility to take a sufficient number of samples of the synthetic fibre rope for testing to establish the synthetic fibre rope properties. Requirements regarding testing are given in DNV-OS-E303 [81]. The change in length testing of the synthetic fibre rope should be defined based on the actual requirements of the mooring design that accurate ULS and FLS may be determined. ABS's guidance notes on the application of synthetic fibre rope for offshore mooring [82] also requires similar testing scope as that of DNV’s.”

Reviewer 2 Report

(1) L31 - I do not agree that near shore is sheltered, it may be most exposed.  Sheltered is in a bay or fetch limited.

(2) Figure 1, sub figures (c) to (f) appear 4 times.

(3) In the introduction, I suggest to clarify more about the classifications of offshore fish farms: Flexible submerged Grid array, Flexible submerged Line array – SPM, Rigid Structures; and to present more references of Flexible – Submerged concepts.

(4) It is important to indicate for each type and reference, weather it is just a theoretical concept or an experienced type. For experienced fish farms, it is valuable to indicate the environmental design conditions and operational experience.

(5) L89 - Hydrodynamic analysis: frequency domain analysis, frequency domain analysis

(6) L305-309, I suggest that all the physical variables are real, such as the time and space dependent potential function (capital Phi), which is the real of the complex space dependent potential function (lowercase phi) times the exponent of minus (iwt).  After solving Phi, all the physical variables are derived as reals.

(7) The Morison’s equation by (4) is for fixed structures, while for floating elements a modified equation exists withe the relative fluid-structure flow.

(8) L588 – Please check “Error! Reference source not found”

(9) The full second order formulation and solution (QTF for all the combinations of sum and difference frequencies) is very complex.  Please provide a proper basic reference for the second order wave-structure theory and solution.

(10) Following the review of classification rules, indicating non-sufficient design practice guidelines, I would expect a conclusion regarding the necessity of rational design approach with direct analysis.

Author Response

Offshore Fish Farms: A review of Standards and Guidelines for Design and Analysis

Y.I Chu, C.M. Wang, H. Zhang, N. Abudussamie, H. Karampour, D.S. Jeng, J. Baumeister, P.A. Aland

The authors wish to thank the editor and the reviewers for their insightful comments and suggestions on our paper. We have revised the paper to address all these comments and suggestions. Below are our point-by-point replies to the reviewers’ comments.

Reviewer #2

(1) L31 - I do not agree that near shore is sheltered, it may be most exposed.  Sheltered is in a bay or fetch limited.

Authors reply: The word “sheltered” has been removed in this paper to have a consistent term of nearshore as opposed to offshore.

(2) Figure 1, sub figures (c) to (f) appear 4 times.

Authors reply: Figure 1 is removed to avoid any copyright issues. Instead, references are added to allow the reader to find the figures.

(3) In the introduction, I suggest to clarify more about the classifications of offshore fish farms: Flexible submerged Grid array, Flexible submerged Line array – SPM, Rigid Structures; and to present more references of Flexible – Submerged concepts.

Authors reply: As this paper mainly focuses on standards and guidelines for design and analysis of offshore fish farms, it may not be so relevant to include details of classification of offshore fish pen designs. Thus, authors removed examples and their photos (see also comment 2 reply). Instead, reference papers are introduced in the introduction section as: “More information and photos of the various offshore fish pen designs and recent developments can be found in a review paper written by Chu et al. [6] and a book chapter written by Wang et al. [9]”

[6] Chu, Y.I.; Wang, C.M.; Park, J.C.; Lader, P.F. Review of Cage and Containment Tank Designs for Offshore Fish Farming. Aquaculture 2020, 519, 734928.

[9] Wang, C.M.; Chu, Y..; Baumeister, J.; Zhang, H.; Jeng, D.S.; Abdussamie, N. Offshore Fish Farming: Challenges and Developments in Fish Pen Designs. In Global Blue Economy: Analysis, Developments, and Challenges; CRC Press, 2022.

 (4) It is important to indicate for each type and reference, weather it is just a theoretical concept or an experienced type. For experienced fish farms, it is valuable to indicate the environmental design conditions and operational experience.

Authors reply: As mentioned in replies 2 and 3, there are other review papers that provide theoretical and experimental design conditions and operational experience for various fish pen designs. This paper keeps the focus on standards and guidelines.

(5) L89 - Hydrodynamic analysis: frequency domain analysis, frequency domain analysis

Authors’ reply: The word “frequency” in the last named analysis is replaced by the word “time”.

(6) L305-309, I suggest that all the physical variables are real, such as the time and space dependent potential function (capital Phi), which is the real of the complex space dependent potential function (lowercase phi) times the exponent of minus (iwt).  After solving Phi, all the physical variables are derived as reals.

Authors’ reply: Equation (1) is revised to show the real part of potential function to derive resultant physical quantities.

(7) The Morison’s equation by (4) is for fixed structures, while for floating elements a modified equation exists withe the relative fluid-structure flow.

Authors’ reply: Equation (4) is revised to present the modified Morison equation for floating structures.

(8) L588 – Please check “Error! Reference source not found”

Authors’ reply: The error reference is corrected.

(9) The full second order formulation and solution (QTF for all the combinations of sum and difference frequencies) is very complex.  Please provide a proper basic reference for the second order wave-structure theory and solution.

Authors’ reply:  The following basic reference for the second order wave theory and solution is cited,

[44]     Choi, Y.R.; Hong, S.Y.; Choi, H.S. An Analysis of Second-Order Wave Forces on Floating Bodies by Using a Higher-Order Boundary Element Method. Ocean Eng. 2001, 28, 117–138, doi:10.1016/S0029-8018(99)00064-5.

(10) Following the review of classification rules, indicating non-sufficient design practice guidelines, I would expect a conclusion regarding the necessity of rational design approach with direct analysis.

Authors reply: The concluding remarks includes the reviewer’s point as follows: “More efforts are needed to provide a unified guideline as well as a more rational design approach through direct analyses in designing offshore fish pens in order to gain industry confidence with regard to offshore farms.”

Reviewer 3 Report

The present study entitled "Offshore Fish Farms: A review of Standards and Guidelines for Design and Analysis"
with reference number of "jmse-2258833, has been evaluated" has been evaluated.
Based on scientific and technical quality of the paper, I can say that the paper has been written with fluent English, well prepared and referenced properly.
Hence, I can recommend to accept the paper in its present form.
For your information.
Kind regards

The reviewer

Author Response

Offshore Fish Farms: A review of Standards and Guidelines for Design and Analysis

Y.I Chu, C.M. Wang, H. Zhang, N. Abudussamie, H. Karampour, D.S. Jeng, J. Baumeister, P.A. Aland

The authors wish to thank the editor and the reviewers for their insightful comments and suggestions on our paper. We have revised the paper to address all these comments and suggestions. Below are our point-by-point replies to the reviewers’ comments.

Reviewer #3

he present study entitled "Offshore Fish Farms: A review of Standards and Guidelines for Design and Analysis"
with reference number of "jmse-2258833, has been evaluated" has been evaluated.
Based on scientific and technical quality of the paper, I can say that the paper has been written with fluent English, well prepared and referenced properly.
Hence, I can recommend to accept the paper in its present form.
For your information.
Kind regards
The reviewer

Authors reply: Authors appreciate the reviewer for kind feedback and acceptance to publish this manuscript.

Reviewer 4 Report

 Introduction   You can also refer some of the papers of Professor O.M. Faltinsen: Shen, Yugao; Firoozkoohi, Reza; Greco, Marilena; Faltinsen, Odd Magnus. (2021)

Experimental investigation of a closed vertical cylinder-shaped fish cage in waves. Ocean Engineering. volum 236. Faltinsen, Odd Magnus. (2011)

Hydrodynamic Aspects of a Floating Fish Farm with Circular Collar. Proceedings 26th International Workshop on Water Waves and Floating Bodies.  

1)    First order wave force (Linear wave force) It seems like the equations (1)-(4), are from the user’s manual of ANSYS AQWA. Please give a reference, and the version of ANSYS 2017 (R1 or R2?).  

Line 363: Give a reference for WADAM. Also the open software NEMOH v.3 calculates the QTF’s.  

References:   R. Kurnia, G. Ducrozet, and J.-C. Gilloteaux. Second Order Dfference- and Sum-Frequency Wave Loads in the Open-Source Potential Flow Solver NEMOH. In International Conference on Offshore Mechanics and Arctic Engineering, volume 5A: Ocean Engineering, 06 2022. V05AT06A019, https://doi.org/10.1115/OMAE2022-79163  

3. 2.2. Current You can also include the journal paper of: Mazarakos T.P., and Mavrakos, S.A. (2012), Wave current interaction on a vertical truncated cylinder floating in finite depth waters. Institution of Mechanical Engineers, Part M: Journal of Engineering for the Maritime Environment Vol. 227 (3): 243 - 255, ISSN 1475-0902, as a reference.  

Lines 714-715: Give references for: AquaSIM, WAMIT and SIMO.  

Give a reference for Eqs. (18) and (19).  

General comments   Please check once again all the manuscript because: “Error! Reference source not found”, and it is very difficult for the reviewer to check the References.

Author Response

Offshore Fish Farms: A review of Standards and Guidelines for Design and Analysis

Y.I Chu, C.M. Wang, H. Zhang, N. Abudussamie, H. Karampour, D.S. Jeng, J. Baumeister, P.A. Aland

The authors wish to thank the editor and the reviewers for their insightful comments and suggestions on our paper. We have revised the paper to address all these comments and suggestions. Below are our point-by-point replies to the reviewers’ comments.

Reviewer#4

Introduction   You can also refer some of the papers of Professor O.M. Faltinsen: Shen, Yugao; Firoozkoohi, Reza; Greco, Marilena; Faltinsen, Odd Magnus. (2021)

Experimental investigation of a closed vertical cylinder-shaped fish cage in waves. Ocean Engineering. volum 236. Faltinsen, Odd Magnus. (2011)

Hydrodynamic Aspects of a Floating Fish Farm with Circular Collar. Proceedings 26th International Workshop on Water Waves and Floating Bodies.  

Authors’ reply: Thank you for introducing the interesting papers. The revision cited those papers in Section 3.4 where the sloshing impact was mentioned.

1)    First order wave force (Linear wave force) It seems like the equations (1)-(4), are from the user’s manual of ANSYS AQWA. Please give a reference, and the version of ANSYS 2017 (R1 or R2?).  

Authors’ reply: The reference for the equations (1)-(4) is included. Please be informed that the cited reference is ANSYS theory manual 2017 which is a theoretical basis for both R1 and R2 versions.

Line 363: Give a reference for WADAM. Also the open software NEMOH v.3 calculates the QTF’s.

References:   R. Kurnia, G. Ducrozet, and J.-C. Gilloteaux. Second Order Dfference- and Sum-Frequency Wave Loads in the Open-Source Potential Flow Solver NEMOH. In International Conference on Offshore Mechanics and Arctic Engineering, volume 5A: Ocean Engineering, 06 2022. V05AT06A019, https://doi.org/10.1115/OMAE2022-79163  

Authors’ reply: The reference for WADAM is included, and NEMOH v.3 is included as one of examples of software packages for QTF calculations.

  1. 2.2. Current You can also include the journal paper of: Mazarakos T.P., and Mavrakos, S.A. (2012), Wave current interaction on a vertical truncated cylinder floating in finite depth waters. Institution of Mechanical Engineers, Part M: Journal of Engineering for the Maritime Environment Vol. 227 (3): 243 - 255, ISSN 1475-0902, as a reference.

Authors’ reply: The recommended reference is added in Section 3.2, Design environment loads.

Lines 714-715: Give references for: AquaSIM, WAMIT and SIMO.  

Give a reference for Eqs. (18) and (19).  

Authors’ reply: References are included in the mentioned software packages and equations.

General comments   Please check once again all the manuscript because: “Error! Reference source not found”, and it is very difficult for the reviewer to check the References.

Authors’ reply: The error references has been corrected in the revised manuscript.

Round 2

Reviewer 2 Report

No additional comments.